# Characterization and Dye Decolorization Potential of Two Laccases from the Marine-Derived Fungus *Pestalotiopsis* sp.

**DOI:** 10.3390/ijms20081864

**Published:** 2019-04-15

**Authors:** Saowanee Wikee, Juliette Hatton, Annick Turbé-Doan, Yann Mathieu, Marianne Daou, Anne Lomascolo, Abhishek Kumar, Saisamorn Lumyong, Giuliano Sciara, Craig B. Faulds, Eric Record

**Affiliations:** 1Department of Biology, Faculty of Science, Chiang Mai University, Chiang Mai 50200, Thailand; wikeemammaam@gmail.com (S.W.); scboi009@gmail.com (S.L.); 2Center of Excellence in Microbial Diversity and Sustainable Utilization, Chiang Mai University, Chiang Mai 50200, Thailand; 3Sup’Biotech, Villejuif, 94800 Paris, France; hatton.juliette@gmail.com; 4INRA UMR1163, Biodiversité et Biotechnologie Fongiques, Aix-Marseille Université, 13288 Marseille, France; annick.doan@univ-amu.fr (A.T.-D.); yann.mathieu85@gmail.com (Y.M.); mariane.daou@Etu.univ-amu.fr (M.D.); anne.lomascolo@univ-amu.fr (A.L.); giuliano.sciara@inra.fr (G.S.); craig.faulds@univ-amu.fr (C.B.F.); 5Department of Genetics & Molecular Biology in Botany, Institute of Botany, Christian-Albrechts-University, 24098 Kiel, Germany; akumar@bot.uni-kiel.de

**Keywords:** laccase, *Pestalotiopsis*, salt tolerance, dye decolorization, heterologous expression.

## Abstract

Two laccase-encoding genes from the marine-derived fungus *Pestalotiopsis* sp. have been cloned in *Aspergillus niger* for heterologous production, and the recombinant enzymes have been characterized to study their physicochemical properties, their ability to decolorize textile dyes for potential biotechnological applications, and their activity in the presence of sea salt. The optimal pH and temperature of *Ps*Lac1 and *Ps*Lac2 differed in relation to the substrates tested, and both enzymes were shown to be extremely stable at temperatures up to 50 °C, retaining 100% activity after 3 h at 50 °C. Both enzymes were stable between pH 4–6. Different substrate specificities were exhibited, and the lowest *K_m_* and highest catalytic efficiency values were obtained against syringaldazine and 2,6-dimethoxyphenol (DMP) for *Ps*Lac1 and *Ps*Lac2, respectively. The industrially important dyes—Acid Yellow, Bromo Cresol Purple, Nitrosulfonazo III, and Reactive Black 5—were more efficiently decolorized by *Ps*Lac1 in the presence of the redox mediator 1-hydroxybenzotriazole (HBT). Activities were compared in saline conditions, and *Ps*Lac2 seemed more adapted to the presence of sea salt than *Ps*Lac1. The overall surface charges of the predicted *Ps*Lac three-dimensional models showed large negatively charged surfaces for *Ps*Lac2, as found in proteins for marine organisms, and more balanced solvent exposed charges for *Ps*Lac1, as seen in proteins from terrestrial organisms.

## 1. Introduction

Laccases are well-known biocatalysts that have been the focus of considerable attention for decades due to their ability to oxidize a wide range of substrates [1]. Laccases (benzenediol: oxygen oxidoreductase; EC 1.10.3.2) are multicopper oxidases and are classed as family 1 of the Auxiliary Activities (AA1) in the Carbohydrate Active Enzymes database (http://www.cazy.org/). They oxidize both phenolic and non-phenolic lignocellulosic compounds, as well as a number of chemically related environmental pollutants [2], through the reduction of a molecule of oxygen to water together with related compounds that correspond to many environment pollutants [3]. However, not all substrates are directly oxidized by laccases because of their size, their limited accessibility, or their redox potential. To overcome these limitations, small molecular mass mediators are often used, which play a role as transitional and diffusible elicitors of laccase activity, by reaching parts of the lignin biopolymer that are inaccessible to the laccase polypeptide. Further on, oxidized radical forms of mediators, generated by laccase activity, have the potential to react with high redox-potential target substrates [4]. Laccases are expressed in many basidiomycete and ascomycete fungi, as well as bacteria, plants, and insects. In some fungi, the level of laccase expression is regulated during fruiting body development and sexual differentiation [5,6]. Fungal laccases are commonly known and widely used in industrial applications, such as textile dye bleaching, pulp bleaching, and in environment-friendly applications such as decolorization, bioremediation, detoxification, and waste water treatment [7,8].

Marine-derived fungi are often associated with sponges and cnidarians, and are hypothesized to be the agents responsible for the organic decomposition of biomaterials by these organisms [9]. They play an important role in nutrient regeneration cycles as decomposers of dead and decaying organic matter [10]. The physical factors that influence their ability to adapt and grow in a marine environment are salinity, pH, low water potential, and a high concentration of sodium and chloride ions [11]. In addition, while terrestrial fungi are able to grow at pH 4.5–6.0, facultative marine fungi were demonstrated to grow and produce numerous extracellular enzymes at pH 7–8 [12,13]. Not astonishingly, laccases from terrestrial fungi are typically active at acidic pH [14]. While the fungal degradation of lignocellulose in terrestrial ecosystems has been intensively studied over the last few decades, very little is known about the mechanisms employed by marine-derived fungi to break down this biomass. A few recent studies are available in which genes from marine-derived fungi involved in biomass degradation were analyzed at the genome level [15,16,17,18]. Furthermore, little is known as to how the coded enzymes have adapted to marine conditions, especially to the presence of high concentrations of salt. For instance, it was demonstrated that the marine fungus *Phlebia* sp., when grown in the presence of 3% sea salt, secretes two different manganese peroxidases (MnP) that are distinct from the one produced in the absence of salt, suggesting that marine fungi may possess alternative sets of lignocellulolytic enzymes adapted to growth in different salt conditions [19].

Mangroves are very interesting marine ecosystems that are found on the coasts of tropical and subtropical regions and are composed of specialized plant species [20]. Mangrove fungi are thought to have a key role in the degradation of organic matter, including lignocellulosic biomass [21]. Recently, Arfi et al. [22] described the enzymatic adaptation of a mangrove fungus, *Pestalotiopsis* sp., in the presence of sea salt. Proteomic analysis demonstrated that the profile of secreted lignocellulolytic enzymes undergoes profound changes in salt conditions, such as a significant reduction of oxidases, and the secretion of specific, salt-adapted carbohydrate-active enzymes. Patel et al. [23] produced and characterized two lytic polysaccharide monoxygenases (LPMOs) previously identified in the secretome of *Pestalotiopsis* sp. during growth under saline conditions [22]. One of the two enzymes was still active on cellulose in the presence of 3.5% (*w*/*v*) sea salt. Marine-derived fungi produce a rich set of enzymes [15,16,17,18], and several were already demonstrated to be amenable to industrial and environmental applications [10]. The fungus *Pestalotiopsis* sp. is commonly known to produce secondary compounds and metabolites [24]. This fungus is distributed worldwide, and there are several reports describing this fungus for its ability to produce compounds of interest for medicinal, agricultural, and industrial applications [25]. To date, two *Pestalotiopsis* genomes are publically available. *Pestalotiopsis fici* Steyaert W106-1/CGMCC3.15140 was isolated from branches of the tea bush *Camellia sinensis*, and has been analyzed at the genomic and transcriptomic level [26]. *P. fici* is characterized by an abundant set of carbohydrate-active enzymes with a specific expansion of pectinases that allow the fungus to utilize intercellular nutrients within the host plants in relation to an adaptation to an endophytic lifestyle. *Pestalotiopsis* sp. KF079 was isolated from the Baltic sea mudflats, and was demonstrated to possess a very rich set of lignocellulose-degrading enzymes together with oxidative enzymes [18]. For this reason, genomic data from *Pestalotiopsis* sp. KF079 were used to select and characterize two fungal laccases, and more specifically their behaviors toward sea salt. These two novel laccases were tested for potential applications for dye decolorization.

## 2. Results

### 2.1. Target Selection, Aspergillus Niger Transformation, Screening, and Protein Purification

Among the 17 putative laccase-encoding genes that were annotated from the genome of *Pestalotiopsis* sp. KF079 (available using BioProject accession ID: PRJNA368776), two were selected for heterologous expression in *A. niger*, based (i) on the divergence of the coded proteins (sharing only 40.5% sequence identity), and (ii) on the presence of the coordination sites for type 1, 2, and 3 coppers, as found in known laccases (Figure 1). The full-length genes that encode for these proteins, which are called *Ps*Lac1 and *Ps*Lac2, consist of 1728 bp and 1683 bp (576 and 561 amino acids), respectively. The two *Ps*Lacs share 41% of identity, and *Ps*Lac1 and *Ps*Lac2 share 59% and 68% identity with *Melanocarpus albomyces* (*Ma*Lac) and *Cryphonectria parasitica* (*Cp*Lac), respectively. The calculated molecular weights are 63 kDa and 62 kDa, with a theoretical pI of 6.17 and 4.13, respectively. The theoretical extinction coefficients at 280 nm are 111,645 M^−1^ cm^−1^ and 142,585 M^−1^ cm^−1^ for *Ps*Lac1 and *Ps*Lac2, respectively.

In order to produce the recombinant enzymes, native signal peptide sequences were removed and replaced by the coding sequence of the *A. niger* glucoamylase signal peptide, followed by a KEX2-like cleavage site located in the expression vector pAN52.4. In co-transformation experiments, protoplasts of *A. niger* D15#26 were transformed with both the pAB4-1 plasmid and the pAN52.4 expression vector. Transformants were selected for their ability to grow without uridine supplementation. Approximately 50 prototrophic transformants were obtained per microgram of expression vector. A total of 30 positive clones for each transformation were cultured in standard liquid cultures and checked daily for protein production by SDS-PAGE and for enzymatic activity by spectrophotometric assays. Approximately 15% of the tested transformants exhibited laccase-like activity in the culture medium. No band was visible on SDS-PAGE, and no laccase activity was detected in the negative controls (transformed by pAB4-1 only). The transformant *Ps*Lac1 reached maximum laccase activity (10.02 nkat mL^−1^) on day 11 and 12. The transformant *Ps*Lac2 reached a peak of activity of 15.12 nkat mL^−1^ on day 8 and 9.

In previous experiments, 3–4% of sea-salt was added to culture media so as to reproduce an artificial seawater environment, affecting the growth of marine bacteria or fungi [22,27,28], and the production of proteins from these organisms. Sea salt is composed of sodium chloride, as well as other minerals and salts, according to the product specification from the manufacturer (Sigma-Aldrich). In our experiments, the effect of sea-salt addition on laccase heterologous production was tested by adding 1% to 5% (*w*/*v*) of sea salt to the *A. niger* growth medium (Figure 2). Laccase activities were found in the secretion medium of the transformant *Ps*Lac1 in non-saline growth conditions, with a maximum at day 12. In all the saline conditions tested instead (1 to 5% sea-salt), laccase activities were strongly abolished. For *Ps*Lac2, laccase activities slightly increased during the first days of growth, and activity sharply dropped down at day 10. Interestingly, laccase activity was significantly increased in cultures grown in 5% sea-salt conditions, reaching a maximum at day 8.

The recombinant laccases were purified from the culture media in a three-step procedure: ultrafiltration, anion-exchange chromatography, and size exclusion chromatography (Table 1, Appendix A). Culture media (containing 25 mg and 228 mg of protein per liter of culture medium, for *Ps*Lac1 and *Ps*Lac2, respectively) were concentrated 9-fold and 3.5-fold, respectively, by ultrafiltration through a polyethersulfone membrane (10 kDa molecular mass cut-off), with a resulting purification factor of 1.1 and 5.2-fold. The two *Ps*Lacs samples were subsequently loaded onto a DEAE column with a purification factor of 3.2-fold and 13-fold, and then purified by size exclusion chromatography yielding approximately 10–20 mg of protein. The final recovery yields were 420% and 276%. 

### 2.2. Enzyme Activity and Stability at Different pH and Temperature

Purified *Ps*Lac1 and *Ps*Lac2 showed a different response to temperature depending on the substrate oxidized (Figure 3). The optimum temperature for *Ps*Lac1 was 80 °C on 2,2′-azino-bis(3-ethylbenzothiazoline-6-sulphonic acid) (ABTS) and 2,6-dimethoxyphenol (DMP) and 65 °C on syringaldazine, while the optimum temperature for *Ps*Lac2 was 65 °C on ABTS, 50 °C on DMP, and 55 °C on syringaldazine. *Ps*Lac1 appears to be more active with high temperature, especially on DMP and syringaldazine. Nevertheless, both enzymes showed similar temperature stability (Figure 4), with similar loss-of-activity profiles at temperatures above 60 °C and stability over 4 h at 30 °C and 50 °C. As already demonstrated for other fungal laccases, laccase activity on ABTS tends to increase at acidic pH, while no activity was found at pH 6 and above (Figure 3). On the other hand, enzyme activity decreased abruptly on DMP and syringaldazine at pH values below 4 or above 6. However, while the optimal pH of fungal laccases is within the range 3–4 [1], for both *Ps*Lacs, a broader optimum pH between pH 4–6 was measured. *Ps*Lac1 was stable between pH 3–6 (Figure 4), while at pH 2, only 30% of the enzymatic activity was retained after 100 h of incubation on ABTS as a substrate (Figure 4). On the other hand, *Ps*Lac2 presented a radically different profile between pH 3–6, retaining at 100 h, after a regular decrease, about 40% to 50% of the starting activity. At pH 2, instead, almost all the activity was lost after 25 h of incubation. Altogether, these results suggest that *Ps*Lac1 seems in general more active and stable at lower pH than *Ps*Lac2.

### 2.3. Substrate Specificity and Kinetic Properties

The substrate oxidizing activity of purified laccases was determined under standard assay condition at 30 °C by using various substrates, and relative activity values were compared. Guaiacol and tannic acid were poorly oxidized by either *Ps*Lac1 or *Ps*Lac2, and ferulic and or 3,4-dihydroxyphenylalanine (DOPA) were not oxidized at all. As such, these compounds were excluded from further characterization.

The kinetic parameters were determined for enzymatic activity on ABTS, DMP, syringaldazine, and *O*-dianisine, and the highest catalytic efficiency was found for syringaldazine (82.936 s^−1^ mM^−1^), followed by ABTS and DMP for *Ps*Lac1 (Table 2, Appendix A). The catalytic efficiency on *Ps*Lac2 on these substrates was overall lower: 9.102 s^−1^ mM^−1^ for DMP, and even lower for syringaldazine, *O*-dianisine, and ABTS.

### 2.4. Decolorization of Commercial Dyes

Purified *Ps*Lac1 and *Ps*Lac2 were effective in their ability to decolorize commercial synthetic dyes (Figure 5). The effect of HBT as a mediator for dye decolorization was also tested, and confirmed that the presence of this mediator improves the decolorization process, probably by facilitating electron transfer between laccases and the substrate molecules, as reported for other laccases [29]. Two profiles could be seen with the six industrial dyes used in this study. With four dyes (Azure Blue, Reactive Black 5, Acid Yellow, and Nitrosulfonazo III), 1-hydroxybenzotriazole (HBT) was essential to achieve 70 to 100% decolorization with *Ps*Lac1after 24 h of incubation. Nevertheless, 20% and 40% decolorization of Acid Yellow 17 and Reactive Black was still achieved with *Ps*Lac1 without HBT after 48 h of incubation. *Ps*Lac2 had similar activity on Reactive Black 5, but was in general much less active on these compounds, and particularly on Azure Blue decolorization, achieving only 20% of decolorization in the presence of HBT after 48 h. Finally, for these compounds finally, no decolorization was observed with *Ps*Lac2 without HBT. The activity of *Ps*Lac1 and *Ps*Lac2 on the two other dyes, Poly-R478 and Bromo Cresol Purple, was pretty independent of HBT, and although *Ps*Lac1 decolorization of Poly-R478 was faster in the presence of the mediator, the same reaction yields were obtained with and without HBT after 48 h, with 50% dye decolorization. For Bromo Cresol Purple, more than 80% of decolorization was achieved within the first 2 h with both laccases, and irrespective of HBT addition.

### 2.5. Influence of Sea Salt on Laccase Activity

In our standard assay, the activity of purified *Ps*Lacs was greatly enhanced by sea-salt addition up to 5% (Figure 6). Salt-mediated activation was particularly striking for *Ps*Lac2, with a sharp activity enhancement at 1%, followed by a slight and steady activity increase, up to 360% with 5% sea salt. Overall, *Ps*Lac1 activity was also enhanced in saline conditions, but to a lesser extent.

### 2.6. Surface Charges of Fungal Laccases

Three-dimensional models of *Ps*Lac1 and *Ps*Lac2 were generated, and the overall surface charges were compared to those of two laccases from terrestrial fungi: the white-rot fungus *Pycnoporus cinnabarinus*, and the ascomycete fungus *Myceliophthora thermophila* (Figure 7). *Pc*Lac (pdb: 2XYB) is part of a large set of laccases (5) deployed by *P. cinnnabarinus* to efficiently degrade lignin [30], and *Mt*Lac (pdb: 6F5K) is a commercially available enzyme that is often used for biotechnological and industrial applications [31]. *Ps*Lac2 showed a large abundance of negative surface charges compared to *Pc*Lac and *Mt*Lac, whereas *Ps*Lac1 seemed to have a better balance of negative and positive charges, as found for the terrestrial enzymes (Figure 7). In line with these results, analysis of the enzyme primary structure showed that *Ps*Lac2 has a 3.95-times higher recurrence of negatively (D + E) over positively charged (R + K) amino acids. In contrast, laccases from terrestrial fungi possess a lower (D + E)/(R + K) ratio, i.e., 1.55 and 2.27 for *Mt*Lac and *Pc*Lac, respectively. Considering *Ps*Lac1, a slightly lower ratio of 1.20 was found. All together, these results suggest that the charge distribution in *Ps*Lac1 is very similar to what found for terrestrial enzymes, while *Ps*Lac2 has excess negatively charged residues, which is a typical feature of marine enzymes [23,32].

## 3. Discussion

Laccases are enzymes of interest due to their wide substrate specificity and suitability for utilization in applications under the green technology banner, thus making them very attractive enzymes for eco-friendly processes. The species *P. cinnabarinus* has been proposed to be of particular interest as a model organism, as it is the first described filamentous fungus relying on laccases as its major ligninolytic enzymes [33]. The genome of this fungus was obtained recently [30], and shows the putative presence of five laccases (CAZy family AA1) together with a large repertoire of other ligninolytic activities. In comparison, very little is still known about the ligninolytic systems of marine-derived fungi. *Pestalotiopsis* sp. J63 was isolated from the oceanic sediment from the seabed of the East China Sea [28]. These authors reported that this *Pestalotiopsis* species secretes high ligninolytic and laccase activity during growth on agricultural residues, suggesting that marine fungi can utilize terrestrial woody and non-woody biomass that ends up in river, coastal, and oceanic areas as a carbon source. The adaptation of enzymes to salty environments is of great interest for the production of more robust biocatalysts and for adapting existing processes where fresh water is restricted. The development of laccases from such marine sources is of particular interest, as many industrial and biotechnological processes relying on laccase activity require enzyme tolerance to heat, ionic strength, and neutral or alkaline pH, which are properties that characterize the environments where marine organisms are able to grow.

To date, a few studies are available that focus on the production and biochemical characterization of laccases obtained from marine organisms [34], with a particular emphasis toward low-redox potential bacterial laccases, such as CotA enzymes from *Bacillus clausii* and *Bacillus subtilis* [35], PPO1 from *Marinomonas mediterranea* [36], and Lbh1 from *Bacillus halodurans* C-125 [37]. In the present work, we aim at providing new data and insight on halotolerant fungal laccases, in order to unveil novel potential enzymatic tools for industrial purposes. In fact, to our knowledge, only two laccases have been characterized to date from an ascomycete and a basidiomycete isolated from mangroves [38,39], and one only have been characterized from a basidiomycete isolated from a sponge in a coastal environment [40]. In a previous work, *Pestalotiopsis* sp. KF079 was isolated from Baltic sea sponges, and its genome was sequenced [18]. In the present study, we browsed this genome and selected two out of 17 putative laccase-encoding genes in order to study the biochemical properties of the coded enzymes and their behavior in sea salt. The two recombinant enzymes, *Ps*Lac1 and *Ps*Lac2, were produced and characterized for their main physicochemical properties. The two laccases from *Pestalotiopsis* sp. KF079 exhibited a high optimal temperature between 55–80 °C, depending on the substrate used. This is similar to what has been reported for laccases from terrestrial fungi [1], such as *Marasmius quercophilus* (optimal temperature 80 °C) [41] and *P. cinnabarinus* (65 °C) [42], or from the marine-derived fungi *Trematosphaeria mangrovei* (65 °C) [38] and *Cerrena unicolor* (70 °C) [39]. Laccase thermal stability can vary a lot, with the half-life time at 50 °C ranging from minutes to hours [1]. Both *Ps*Lac1 and *Ps*Lac2 lose activity after 1 h of incubation at 60 °C, with a half-life time of about 40 min, while at 30 °C and 50 °C, both enzymes were found to be stable for 4 h. This is comparable to the behavior of the laccase from *T. mangrovei* [38], which has a half-life time of 45 min at 50 °C, while the thermostable laccase of *Melanocarpus albomyces* has a half-life time of 5 h [43]. The thermal stability of enzymes has been proposed to rely on several factors, such as the presence of hydrophobic cores or charged residues [44,45]. However, to our knowledge, no clear relationship has been established between these factors and thermal stability in the case of enzymes from marine-derived fungi, which is typically characterized by large negatively charged solvent-exposed surfaces.

Considering the behavior at different pH, fungal laccases exhibit relatively acidic pH optima, although with different tendencies depending on the enzymatic substrate and the underlying oxidation mechanism [46]. For *Ps*Lacs, ABTS was oxidized at a pH lower than 4.0, suggesting the fast oxidation of ABTS to the corresponding radical cation ABTS^+^ at low pH, as previously described for the laccase from the basidiomycete *P. cinnabarinus* [42], while DMP, guaiacol, and syringaldazine were oxidized in the pH range of 4.0 to 6.0. Similar results were obtained for the laccase from the marine fungi *Peniophora sp.* (pH 5.0) [40] and from *C. unicolor* (pH 3.0) [39]. As such, although this was expected for marine-derived enzymes, none of these laccases, including *Ps*Lacs, present an optimum pH in neutral or alkaline conditions. For pH stability, laccases produced by terrestrial fungi are mainly stable at acidic pH [1], and we observed the same for *Ps*Lac1, which showed 30% residual activity after 100 h of incubation at pH 2.0, suggesting similar properties for *Ps*Lac1 and terrestrial active laccases. *Ps*Lac2 instead was more sensitive to low pH; its stability decreased regularly with time between pH 3.0 to 6.0 and abruptly at pH 2.0, where no activity was left after 40 min of incubation. Altogether, these results suggest that *Ps*Lac1 has generally higher activity and stability at lower pH than *Ps*Lac2, which is comparable to the preference for acidic pH described for laccases from terrestrial fungi [14].

In order to gain insight into the enzymatic mechanism of *Ps*Lacs, substrate specificity was tested for non-natural substrates, such as ABTS and some phenolic compounds. Both laccases were pretty active on the low redox substrate ABTS, but the preferred substrates were syringaldazine and DMP for *Ps*Lac1 and *Ps*Lac2, respectively. Tannic acid and guaiacol were the less oxidized substrates, and ferulic acid and DOPA not at all oxidized. In the literature, laccase activity has been tested on a wide range of substrates to determine substrate specificity, and the data show that syringaldazine and ABTS are by far, with some exceptions, the preferred substrates, while for the other substrates, specificity is very variable [1]. In general, laccases share high catalytic constant, as well as a high affinity for syringaldazine (tens of µM) and ABTS (hundreds of µM), whereas the oxidation of other phenolic compounds is considerably slower, and the respective *K_m_* constants are considerably higher [1,10]. *Ps*Lac1 showed similar properties to other laccases, with *K_m_* values of 4 µM and 24 µM for syringaldazine and ABTS, respectively, pointing to high affinities for these substrates. For *Ps*Lac2, instead, higher *K_m_* values (hundreds of µM) point to lower affinities toward syringaldazine and ABTS. For comparison, the two marine-derived fungal laccases from *T. mangrovei* [38] and *C. unicolor* [39] revealed different behaviors with *K_m_* of 1.42 mM and 0.054 mM against ABTS. Considering the catalytic constant, *k_cat_* values of fungal laccases are highly variable depending on the enzymes and the substrates used, typically in the range of 1 to 50 s^−1^ for syringaldazine. For *Ps*Lacs, *k_cat_* for syringaldazine are in the low value range, as has been reported for the laccase from the ascomycete fungus *Botrytis cinerea* [47]. Notably, *Ps*Lac1 exhibited the highest catalytic efficiency for syringaldazine, as its affinity for this substrate is relatively high compared to other fungal laccases. Its *K_cat_*/*K_m_* for ABTS and DMP is still rather high, but it is also 10^3^ times smaller for *o*-dianisidine. In contrast, *Ps*Lac2 turned out to be more versatile, showing lower and comparable catalytic efficiencies for the oxidation of these four compounds, but a higher spectrum of accepted substrates, including *o*-dianisidine. We can conclude that *Ps*Lac1 have suitable properties for the oxidation of phenolic compounds in the (salt-free) conditions tested, whereas *Ps*Lac2 might prove active on a larger number of phenolic compounds, which is a possibility that is worth exploring.

Decolorization potential is another criterion of interest in the study of laccases with respect to the treatment of industrial effluent treatment [48]. Enzymes from marine-derived fungi can be applied to treat textile effluents, which are often characterized by alkaline pH and high ionic strength [11]. Laccase-mediated azo dye decolorization has been well described with crude and purified forms from many fungi; however, most of the laccases required the addition of redox mediators [1,48,49]. The mechanism of action of the laccase-mediator system has been extensively studied in many applications [50,51]. Both bacteria and fungi, such as *Pseudomonas putida* [52], *Trametes hirsute*, *Pleurotus florida* [53], and *Pestalotiopsis guepini* [54] have been tested for their dye-decolorizing ability against synthetic dyes including Blue CA, Black B133, Corazol violet SR, or Crystal Violet. Azo dyes, with some exceptions, seem relatively recalcitrant to degradation, and the nature and position of substituents on their aromatic rings can markedly influence the decolorization efficiency of the laccase. It has been previously reported that immobilized enzymes from *Pestalotiopsis* sp. NG007 are able to decolorize three textile dyes with increasing effect, i.e., Reactive Remazol Navy 4 < Reactive Remazol Violet 9 < Lefavix Blue 16 [55]. In that work, most of the dye decolorization was mainly due to laccase activity and this feature was confirmed with marine-derived strains [11]. Concerning purified enzymes, a laccase from *Polyporus brumalis* showed effective decolorization of a dye, Remazol Brilliant Blue R (RBBR), without any laccase mediator [56]. In the case of *Ps*Lacs, both of them have decolorization activity against synthetic dyes such as Azure Blue, Reactive Black 5, Acid Yellow 17, Nitrosulfonazo III, Poly-R478, and Bromo Cresol Purple in the presence of mediators. Nevertheless, comparable or residual activities were also measured without HBT for Bromo Cresol Purple Poly-R478, Reactive Black 5, and Acid Yellow 17. Our results show that *Ps*Lacs have a potential for dye decolorization without the need for mediators, and further in-depth characterization will be necessary to assess this possibility.

To complete enzyme characterization, we decided to test the behavior of *Ps*Lacs in saline conditions. In fact, heterologous protein expression showed strong salt dependence and a marked difference in the expression profile for the two laccases. Precisely, the production of *Ps*Lac1 and *Ps*Lac2 was highly inhibited and highly boosted, respectively, by the addition of sea salt to the culture media of *A. niger*. The activity of *Ps*Lacs was also affected by sea salt: both enzymes were more active in the presence of salt, although activity enhancement was clearly stronger for *Ps*Lac2. Whereas salt dependence of bacterial low-redox potential laccases has been established for different model systems, very little is still known for marine-derived fungal laccases. One of the few examples is the laccase from *T. mangrovei*, which lost about 50% of its activity in the presence of 1 mM of NaCl [38]. On the contrary, several bacterial low-redox potential laccases have been shown to be resistant and even activated by salts, with striking examples such as the PPO1 laccase from *M. mediterranea*, which is active in up to 1 M of NaCl [36], and the Lbh1 laccase from *B. halodurans* C-125, which is activated by up to 450 mM of salt [37]. As such, the activity profile of *Ps*Lacs in sea salt is a promising step toward the discovery of novel halotolerant biocatalysts, and a property worth investigating in further studies. Salt-adapted enzymes are characterized by highly negative surface charges that are thought to contribute to protein stability and activity in extreme osmolytic conditions [32,57,58]. As seen in our results, *Ps*Lac1 has a low (D + E)/(R + K) ratio, which is even lower than the one found for the two reference laccases from terrestrial fungi used in this study, whereas *Ps*Lac2 has a high ratio of acid/basic residues that is more similar to what is expected for marine enzymes. This observation is comforted by comparing the homology-guided three-dimensional models generated for *Ps*Lacs and the three-dimensional structures of laccases from two terrestrial fungi, the ascomycete *M. thermophila* and the basidiomycete *P. cinnabarinus*. The surface charge distribution of *Ps*Lac1 is in fact reminiscent of those of terrestrial laccases, whereas *Ps*Lac2 shows highly negative solvent exposed surfaces, such as those usually found in marine enzymes. Our previous analysis of surface charges of two lytic polysaccharide monooxygenases (LPMOs) isolated from *Pestalotiopsis sp.* NCi6 showed a similar pattern, with (D+E)/(R+K) ratios of 4.8 and 4.3 respectively, which were comparable to those found for *Ps*Lac2 (3.95), and significantly higher than what has been observed for 21 terrestrial LPMOs from thermophilic and mesophilic fungi (D+E)/(R+K) (between 0.9–2.7) [23].

## 4. Materials and Methods

### 4.1. Strains and Culture Conditions

*Escherichia coli* strain JM109 (Promega, Charbonnieres, France) was used for vector storage and propagation. *Aspergillus niger* strain D15#26 (*pyr*G deficient) [59] was used for the heterologous expression of the laccase-encoding synthetic genes. After co-transformation with two vectors containing the *pyr*G gene and the expression cassette respectively, transformants of *A. niger* were grown for selection on solid minimal medium without uridine and containing 70 mM of NaNO_3_, 7 mM of KCl, 11 mM of KH_2_HPO_4_, 2 mM of MgSO_4_, glucose 1% (*w*/*v*), and trace elements (1000× stock; 76 mM of ZnSO_4_, 178 mM of H_3_BO_3_, 25 mM of MnCl_2_, 18 mM of FeSO_4_, 7.1 mM of CoCl_2_, 6.4 mM of CuSO_4_, 6.2 mM of Na_2_MoO_4_, and 174 mM of EDTA—ethylenediamine tetra-acetic acid). For the screening procedure of the positive transformants, 100 mL of culture medium containing 70 mM of NaNO_3_, 7 mM of KCl, 200 mM of Na_2_HPO_4_, 2 mM of MgSO_4_, glucose 5% (*w*/*v*), and trace elements were inoculated with 2 × 10^6^ spores mL^−^^1^ in a 250-mL baffled flask. *Pestalotiopsis sp.* KF079 is available at the culture collection of the Kiel Center for marine natural products at GEOMAR, Helmholtz Centre for Ocean Research, (Kiel, Germany).

### 4.2. Cloning and Expression of PsLac-Encoding Genes

The open reading frames sequences encoding each *Ps*Lac*s* were synthesized and codon optimized for *A. niger* (GeneArt, Regensburg, Germany), with some modifications. The amino acids of the signal peptides, which were predicted with the program SignalP hosted on the ExPASy Proteomics server (http://www.expasy.ch), were replaced by the 24-amino-acid glucoamylase (GLA) preprosequence from *A. niger* (MGFRSLLALSGLVCNGLANVISKR). Two restriction sites (*Nco*I and *Hind*III) were added at the 5′ and 3′ ends of the sequence, respectively, for cloning into the expression vector pAN52.4 (GenBank/EMBL accession number Z32699). In the final expression cassette, the *Aspergillus nidulans* glyceraldehyde-3-phosphate dehydrogenase-encoding gene (*gpdA*) promoter, the 5′ untranslated region of the *gpdA* mRNA, and the *A. nidulans trpC* terminator were used to drive the expression of the inserted coding sequences.

### 4.3. Transformation, Screening of Transformants, and Laccase Activity Assay

For each recombinant protein, co-transformation was carried out as described by Punt and van den Hondel [60] using both the expression vector containing the expression cassette and pAB4-1 [61] containing the *pyrG* selection marker in a 10:1 ratio. Transformants were selected for uridine prototrophy by growth on selective solid minimal medium (without uridine). In order to screen enzyme production in liquid medium, 100 mL of culture minimal medium (adjusted to pH 5.5 with 1 M of citric acid) were inoculated with 2 × 10^6^ spores mL^−^^1^ in a 250-mL flask. The culture was monitored for 14 days at 30 °C in a shaker incubator (130 rpm), and pH was adjusted daily to 5.5 with 1 M of citric acid. From these liquid cultures, aliquots (2 mL) were collected daily, and cells were pelleted by centrifugation (20 min at 15,000× *g*). The resulting supernatant was concentrated onto microcentrifuge units (Centricon, Merck Millipore, Darmstadt, Germany) with a 30-kDa cut-off, and protein production was confirmed by 12% (*w*/*v*) sodium dodecyl sulfate-polyacrylamide gel electrophoresis and a laccase activity test. Laccase activity was assayed spectrophotometrically by monitoring the oxidization of 0.5 mM of 2,2′-azino-bis (3-ethylbenzothiazoline-6-sulphonic acid) (ABTS) (ε = 3.6.10^4^ M^−1^ cm^−1^) at 420 nm in 20 mM of tartrate buffer pH 4. The reaction was monitored for 1 min at 30 °C in an Uvikon XS spectrophotometer (BioTek Instruments, Colmar, France). Activity is expressed in nkat mL^−1^, i.e., the amount of laccase that oxidizes 1 µmol of substrate per second. Measurements in all the experiments were performed in triplicate, and means and standard deviations were determined.

### 4.4. Influence of Salt on Laccase Production

Two transformants producing the highest *Ps*Lac1 and *Ps*Lac2 laccase activity were cultured in liquid media in the presence of different concentrations of artificial sea salts powder (0%, 1%, 3%, and 5% (*w*/*v*) (Sigma-Aldrich, St. Louis, MO, USA) in order to test the influence of sea salt on laccase production. First, 100 mL of culture medium was inoculated with 2 × 10^6^ spores mL^−1^ in a 500-mL baffled flask at 30 °C in a shaker incubator (110 rpm), and pH was adjusted daily to 5.5 by the addition of 1 M of citric acid. An aliquot (1 mL) was collected each day.

### 4.5. Purification of Recombinant Laccases

For protein production, one positive clone was selected for each expression cassette and cultured without or with 5% sea salt for *Ps*Lac1 and *Ps*Lac2, respectively. Then, 1800 mL (*Ps*Lac1) and 700 mL (*Ps*Lac2) of culture media were prepared as previously described (see Section 4) for large-scale protein production. The cultures were harvested after 11 days (*Ps*Lac1) and 8 days (*Ps*Lac2). Culture media was clarified by filtration through GF/D and GF/F glass fiber filters (Whatman, Maidstone, UK), followed by filtrations through 0.45-μm and 0.22-μm polyethersulfone membranes (Express Plus, Merck Millipore). The collected filtrate was concentrated 10-fold by ultrafiltration through a polyethersulfone membrane with a 10-kDa molecular mass cut-off (Vivaflow crossflow cassette, Sartorius, Les Ulis, France). The supernatant was purified on a diethylaminoethyl (DEAE)-Sepharose fast flow (Amersham Bioscience Europe GmbH, Saclay, Orsay, France) using an AKTA purifier (GE Healthcare Life Science, Velizy-Villacoublay, France). The sample was loaded onto the column equilibrated with a binding buffer (50 mM of phosphate buffer pH 7, containing 25 mM of NaCl), using a linear gradient of 0–50% of the elution buffer (50 mM of phosphate buffer pH 7, containing 1M of NaCl). Activity was determined in the collected fractions, and protein production was evaluated by SDS–PAGE. The active fractions were concentrated through a 10-kDa molecular mass cut-off Amicon membrane (Millipore). The concentrated samples were loaded onto a Sephacryl S-200HR size exclusion chromatography column (GE Healthcare Life Science) equilibrated in 50 mM of sodium acetate buffer pH 5, containing 50 mM of NaCl. Five-mL fractions were collected and assayed for laccase activity as described above, and protein production was tested by SDS-PAGE. For the different purification steps, the total protein concentration was determined with the Bradford assay using the BioRad Protein Assay Kit (BioRad, Marnes-la-Coquette, France) and bovine serum albumin (BSA) as a standard. Purified laccase was dialyzed against 10 mM of tartrate buffer, and final protein concentration was determined spectrophotometrically at 280 nm using a NanoDrop 2000 (Thermo Fisher Scientific, Illkirch, France), and the respective theoretical molar extinction coefficients at 280 nm, 111,645 M^−1^ cm^−1^ (*Ps*Lac1), and 142,585 M^−1^ cm^−1^ (*Ps*Lac2).

### 4.6. Bioinformatic Analysis

The molecular mass, theoretical pI, and molar extinction coefficient of enzymes were predicted by the ProtParam tool (http://web.expasy.org/protparam/). Protein sequences were aligned using MUSCLE [62] and CLUSTAL W [63] at http://www.ebi.ac.uk/Tools/msa/. Signal peptides were predicted using Signal P [64] at http://www.cbs.dtu.dk/services/SignalP/. The automated protein structure homology modeling online tool Phyre 2 [65] was used to predict the three-dimensional models of *Ps*Lac1 and *Ps*Lac2 using the closest homologs of known structure available in the Protein Data Bank (PDB). For instance, c2q9oA (*Melanocarpus albomyces* laccase, 59% identity) and c3ppsD (*Thielavia arenaria* laccase, 59% identity) were used for *Ps*Lac1. Protein models c2q9oA (*Melanocarpus albomyces* laccase, 40% identity) and c3sqrA (*Botrytis aclada* laccase, 49% identity) were used as references for *Ps*Lac2. Surfaces charges were calculated, and all the figures were prepared with PyMOL.

### 4.7. Influence of Temperature and pH

To determine optimal temperature, laccases were incubated over a temperature range from 20 to 80 °C, and the activity was determined at the different temperatures in standard conditions. For the pH profiles, laccase activity was determined in 200 mM of tartrate buffer from pH 2 to 6 and phosphate buffer at pH 7 against ABTS, DMP, and syringaldazine as substrates at 30 °C.

To define the thermal stability, laccases were incubated at different temperatures (30 to 80 °C) for 15, 30, 60, 120, and 240 min. Thermal inactivation was stopped by immediately cooling the treated protein aliquot on ice, and activity was measured under standard conditions as described above. The pH stability was determined by incubating laccases in tartrate buffer at different pH (2, 3, 4, 5 and 6) for 4, 24, 48, and 96 h at 30 °C, and then assaying the activity in standard conditions.

In all the conditions, 100% relative activity was fixed in a range of 5 to 20 nkatal per mL of laccase activity.

### 4.8. The Substrates Specificity and Kinetics

The substrate specificity of purified laccases was determined by following spectrophotometric changes in standard conditions at specific wavelengths for each substrate: 2,2′-azino-bis(3-ethylbenzothiazoline-6-sulphonic acid) (ABTS) (420 nm), tannic acid (420 nm), *o-*dianisidine (460 nm), 2,4-dimethylphenol (DMP) (469 nm), guaiacol (470 nm), syringaldazine (530 nm), ferulic acid (420 nm), and 3,4-dihydroxyphenylalanine (DOPA) (475 nm).

Kinetic constants were determined at 30 °C, and either at pH 4 using ABTS as a substrate, or pH 5 for DMP, syringaldazine, and O-dianisidine (substrate concentration range of 0.005 mM to 2 mM for all substrates, Appendix A), using the Lineweaver–Burk double-reciprocal plots in the presence and absence of co-solvent. Values were estimated using a non-linear regression model (GraFit software, Erithacus Sofware, Horley, UK).

### 4.9. Decolorization Properties

The loss of absorbance for the synthetic dyes Acid Yellow 17 (404 nm), Azure Blue (645 nm), Bromo Cresol Purple (431 nm), Nitrosulfonazo III (575 nm), Poly R478 (520 nm), and Reactive Black 5 (600 nm) was used for determining the decolorization properties of laccase in the presence/absence of a redox mediator 1-hydroxybenzotriazole (HBT). The reaction mixture contained laccase (6 U mL^−1^), dye solutions (final concentration: 0.004% *w*/*v*), tartrate buffer (20 mM, pH 4) and 0.9 mM of HBT in a total volume of 1 mL. The decolorization was detected by measuring the decrease of color absorbance with time (1, 2, 4, 24, and 48 h). Heat-inactivated laccases were used as controls. The percentage of decolorization efficiency was calculated as follows:Decolorization (%) = (A_i_ − A_t_)/A_i_ × 100(1)
where A_i_ is initial absorbance of dyes, and A_t_ is the absorbance of dye after each time point [66].

### 4.10. Influence of Sea Salt on Laccase Activity

The influence of sea-salt on laccase activity was assessed by spectrophotometric measurements in standard assay conditions, as described above, after sea-salt addition (1–5% *w*/*v*).

### 4.11. Sequence Information

The gene sequences encoding *Ps*Lac1 and for *Ps*Lac2 were deposited in the GenBank database under the accession codes KY554800 and KY554801, respectively.

## 5. Conclusions

The results presented here have the objective to provide insight on marine-derived enzymes and more specifically on laccases. The two heterologously expressed laccases were demonstrated to possess different biochemical properties, but also different potentials for industrial applications. Future research will be needed in order to explain the enzyme properties in relation to halophilic conditions, especially at the structural level. This knowledge will be of interest to develop more robust enzymes for biotechnological applications.

## Figures and Tables

**Figure 1 ijms-20-01864-f001:**
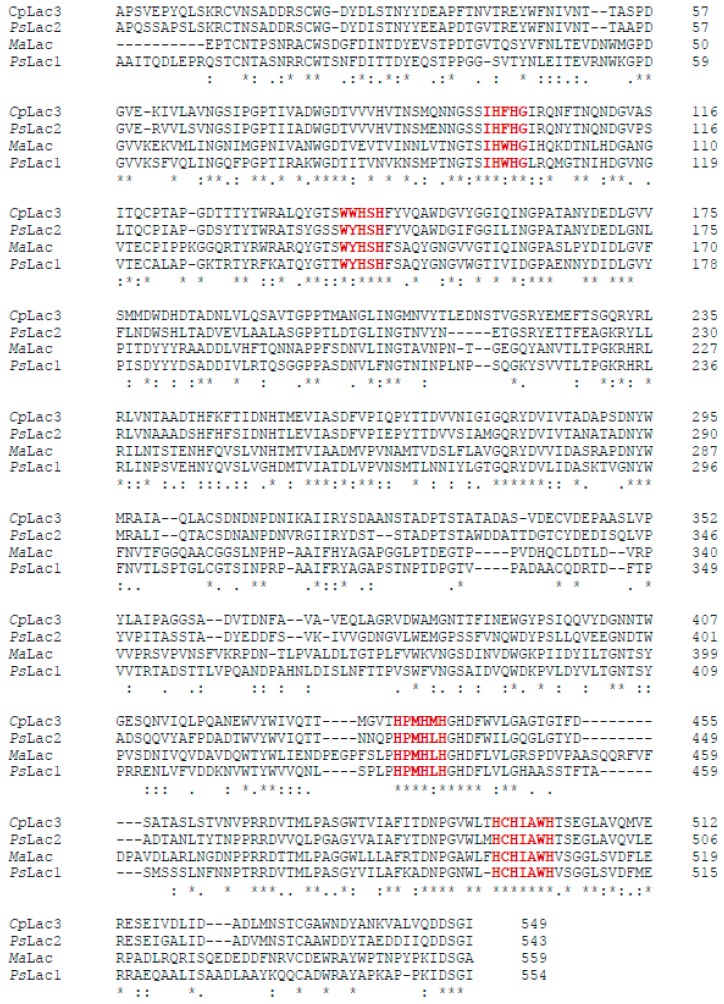
Alignments of the closest characterized laccases from *Melanocarpus albomyces* (*Ma*Lac, 3DKH_A) and *Cryphonectria parasitica* (*Cp*Lac3, AAY99671) to *Pestalotopsis* sp. Lac1 (KY554800), and *Pestalotiopsis* sp. Lac2 (KY554801) using the CLUSTAL W sequence alignment algorithm. Perfect matches are represented by an asterisk, high-amino acid similarities are represented by double dots, and low-amino acid similarities are represented by single dots. Gaps (-) were introduced for maximum alignment. The amino acids involved in the coordination sites for the type 1, 2 and 3 coppers are in red and bold.

**Figure 2 ijms-20-01864-f002:**
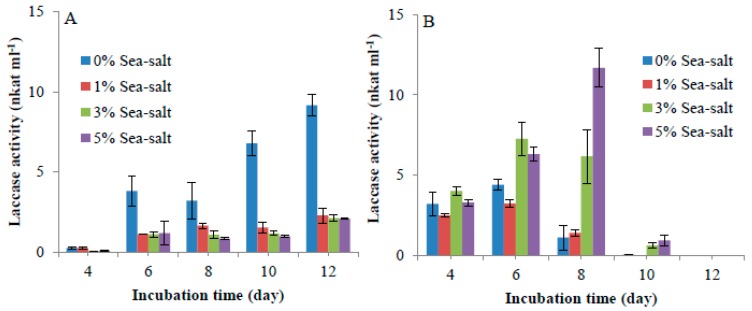
Laccase production in *A. niger* under saline conditions for *Ps*Lac1 (**A**) and *Ps*Lac2 (**B**). Laccase activities were determined on culture medium with 2,2′-azino-bis(3-ethylbenzothiazoline-6-sulphonic acid) (ABTS) as the substrate in standard conditions. Each data point (mean +/− standard deviation) is the result of triplicate experiments.

**Figure 3 ijms-20-01864-f003:**
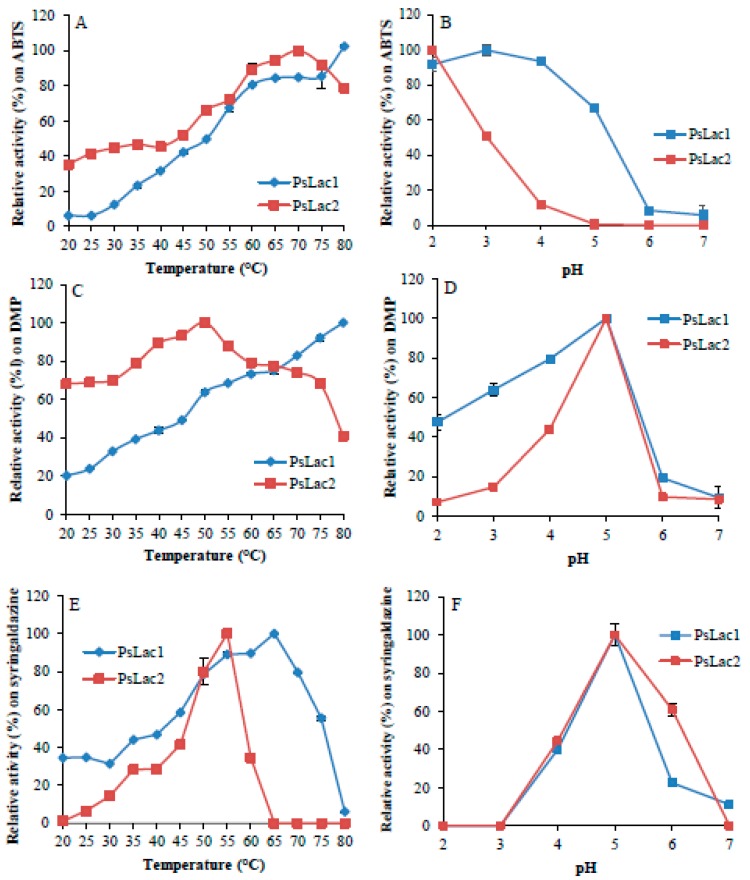
Temperature and pH dependence of *Ps*Lac1 and *Ps*Lac2 activity on different substrates. Values were calculated as a percentage of maximum activity (set to 100%) at optimum temperature and pH. The oxidation of ABTS (**A**,**B**), 2,6-dimethoxyphenol (DMP) (**C**,**D**), and syringaldazine (**E**,**F**) (5 mM each) was determined for the temperature curve in 20 mM of tartrate buffer pH 4.0, and for the pH curve at 30 °C. Each data point (mean +/− standard deviation) is the result of triplicate experiments.

**Figure 4 ijms-20-01864-f004:**
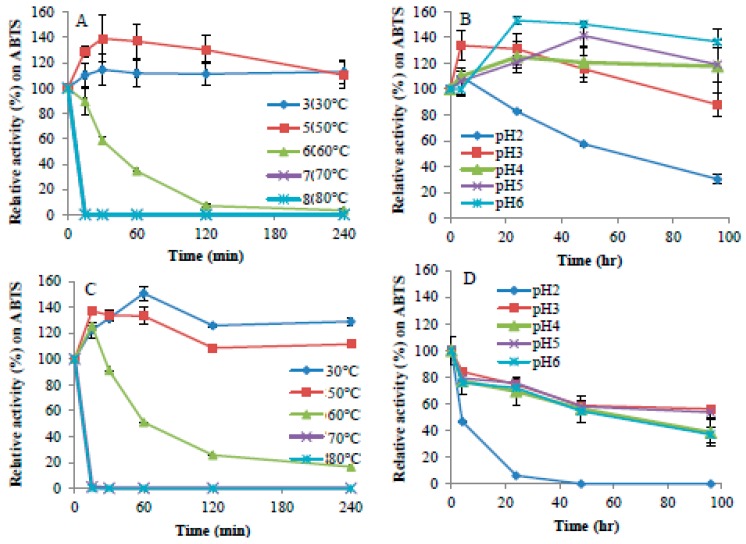
Thermal and pH stabilities of purified *Ps*Lac1 and *Ps*Lac2. Residual activities were estimated after 15, 30, 60, 120, and 240 min of incubation at five different temperatures ranging from 30 to 80 °C (**A**,**C**) or at five different pHs ranging from 2.0 to 6.0 (**B**,**D**) for *Ps*Lac1 (**A**,**B**) and *Ps*Lac2 (**C**,**D**). Residual activities are expressed as a percentage of the initial activity (point at time 0, measured immediately after adding the enzyme), which was set to 100%. Assays were performed using ABTS as a substrate in standard conditions. Each data point (mean +/− standard deviation) is the result of triplicate experiments.

**Figure 5 ijms-20-01864-f005:**
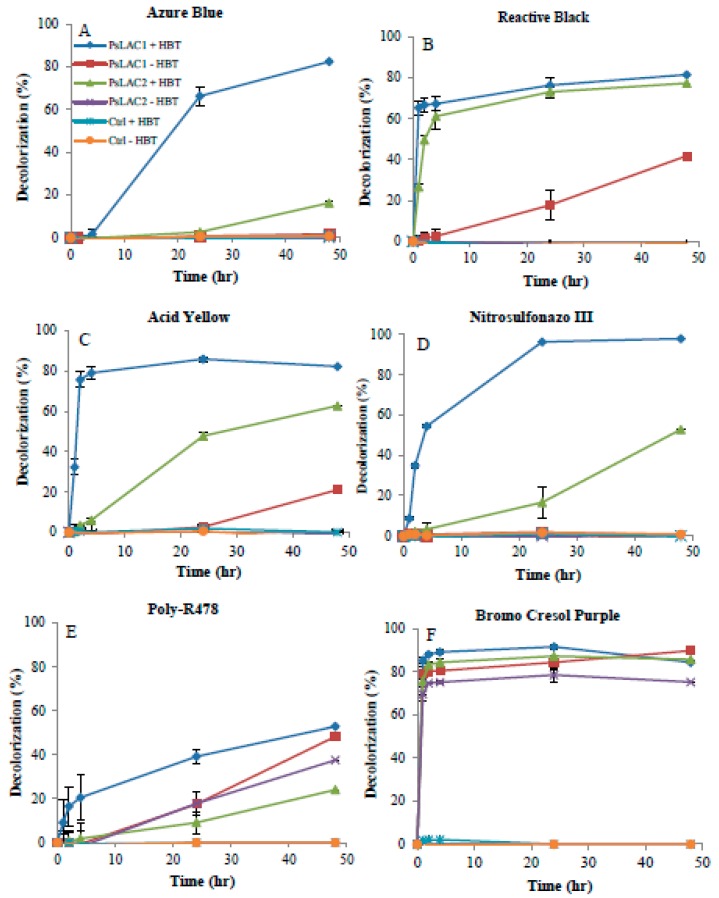
Decolorization of industrial dyes by *Ps*Lac1 and *Ps*Lac2. (**A**–**F**) The decolorization was detected by measuring the decrease of absorbance at specific wavelengths with time (1 h, 2 h, 4 h, 24 h, and 48 h). Each data point (mean +/− standard deviation) is the result of triplicate experiments.

**Figure 6 ijms-20-01864-f006:**
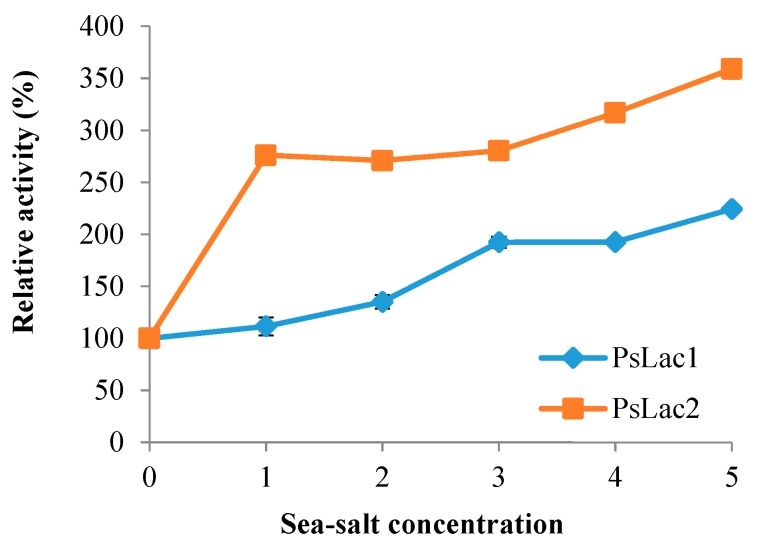
Effect of sea-salt addition (0 to 5% *w*/*v*) on the activity of *Ps*Lac1 and *Ps*Lac2. Assays were performed using ABTS as substrate in standard conditions. Each data point (mean +/− standard deviation) is the result of triplicate experiments.

**Figure 7 ijms-20-01864-f007:**
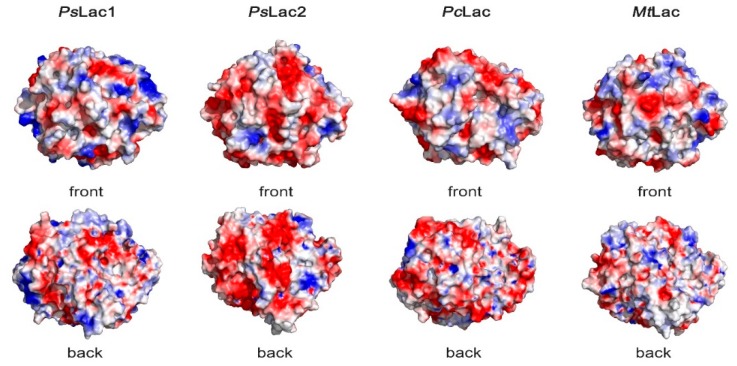
Surface charge plots (negative and positive charges are in red and blue, respectively) of *Ps*Lac1 and *Ps*Lac2 compared to those from *Pycnoporus cinnabarinus* laccase, *Pc*Lac (pdb: 2XYB) and from *Myceliphthora thermophila* laccase, *Mt*Lac (pdb: 6F5K). The surface potentials were calculated using the vacuum electrostatics function of the PyMOL molecular graphics system (Schrödinger, New York, NY, USA).

**Table 1 ijms-20-01864-t001:** Purification steps for each recombinant laccase produced in *Aspergillus niger* D15. DEAE: diethylaminoethyl.

Purification Steps	Volume (mL)		Total Activity (nkat)		Protein (mg)		Specific Activity (nkat mg^−1^)		Activity Yield (%)		Purification Factor (Fold)	
	*Ps*LAC1	*Ps*LAC2	*Ps*LAC1	*Ps*LAC2	*Ps*LAC1	*Ps*LAC2	*Ps*LAC1	*Ps*LAC2	*Ps*LAC1	*Ps*LAC2	*Ps*LAC1	*Ps*LAC2
Culture medium	1800	700	20081	20055	45	160	446	125	100	100	1.0	1.0
Ultrafiltration	200	200	29404	41218	58	63	507	654	146	206	1.1	5.2
DEAE	200	30	88174	54015	62	33	1422	1636	439	269	3.2	13
Gel filtration	2	5	84244	55545	20	12	4212	4629	420	276	9.4	37

**Table 2 ijms-20-01864-t002:** Kinetic parameters of *Ps*Lac and *Ps*Lac2.

Substrate	Structure	*Ps*Lac1			*Ps*Lac2		
		*K_m_* (mM)	*K_cat_* (s^−^^1^)	*K_cat_*/*K_m_* (s^−^^1^ mM^−^^1^)	*K_m_* (mM)	*K_cat_* (s^−^^1^)	*K_cat_*/*K_m_* (s^−^^1^ mM^−^^1^)
ABTS	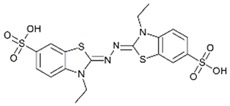	0.024	0.696	29.245	0.100	0.431	4.331
DMP	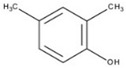	0.100	2.441	24.532	0.020	0.178	9.102
Syringaldazine	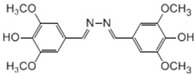	0.004	0.365	82.936	0.101	0.573	5.672
*O*-dianisidine	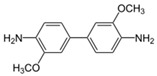	0.647	0.057	0.089	0.144	0.748	5.187

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
