# Peer review of "Characterization and Dye Decolorization Potential of Two Laccases from the Marine-Derived Fungus Pestalotiopsis sp."

_ijms, 2019, doi:10.3390/ijms20081864_

Round 1

Reviewer 1 Report

From my point of view these results could be published after taking care of some minor issues:

Figure 3 was not properly seen, probably due to an error during file conversion?

It would be nice to have a call for supporting information Figures 2 and 3 in section 2.3.

Some typos:

-        Line 277: when it says “…achieved with PsLac1 without ABTS…”, shouldn’t it be “…achieved with PsLac1 without HBT…”?

-        The space between the figure and the units of measurement is missing throughout the manuscript.

-        Line 258: an “a” is missing in “Guaiacol”

-        There are many more typos throughout the manuscript that should be carefully checked and corrected

Author Response

Reviewer 1:

From my point of view these results could be published after taking care of some minor issues:

Figure 3 was not properly seen, probably due to an error during file conversion?

I am sorry for this problem.

It would be nice to have a call for supporting information Figures 2 and 3 in section 2.3.

This was added in Section 2.3 just after the Table 2 reference.

Some typos:

-        Line 277: when it says “…achieved with PsLac1 without ABTS…”, shouldn’t it be “…achieved with PsLac1 without HBT…”?

Yes, thank you for the remark.

-        The space between the figure and the units of measurement is missing throughout the manuscript.

We solved the problem in the revised version

-        Line 258: an “a” is missing in “Guaiacol”

This was corrected in the text.

-        There are many more typos throughout the manuscript that should be carefully checked and corrected

We checked the manuscript and found some typos that were corrected.

Reviewer 2 Report

Most of my comments from the first reviewing round were addressed. However, I would like to point out some new comments that came up in the revised manuscript as well as repeat my previous comments, since they were not commented on from the authors' side in a rebuttal.

The introduced a spacing in between the unit °C and the numbers before. Actually °C is one of the few units (also %) were there should be no spacing in between the unit and the number. I apologize for picky on this.

Figure 1: As already pointed out in my first review report, I still consider this alignment focusing on the Cu binding site only not so useful. It would be much better to show the full sequences and giving the relatedness of those to each other in a table.

p. 7, l. 185: The supplementary material is missing. I hope that it would have been SDS-PAGE data.

p. 7, l. 191-192: The authors state that elimination of inhibiting factors led to increase in purification yield. No! These higher yields in activity are mainly due to removal of contaminating proteins and actual enrichment of the laccase, which can be seen in the increase of specific activity.

Figure 3: As already pointed out in my first review report, there are still not enough tickmarks on the temperature x-axis and information on the absolute activity in nkat units at 100% is still missing.

p. 16, l. 363-366: As already pointed out in my first review report, You cannot state that Pestalotiopsis is secreting the laccases based on the presented results, because you only did recombinant expression in Aspergillus. It was never shown in this study that Pestalotiopsis is secreting these laccases.

Author Response

Reviewer 2:

Most of my comments from the first reviewing round were addressed. However, I would like to point out some new comments that came up in the revised manuscript as well as repeat my previous comments, since they were not commented on from the authors' side in a rebuttal.

 The introduced a spacing in between the unit °C and the numbers before. Actually °C is one of the few units (also %) were there should be no spacing in between the unit and the number. I apologize for picky on this.

This was corrected.

Figure 1: As already pointed out in my first review report, I still consider this alignment focusing on the Cu binding site only not so useful. It would be much better to show the full sequences and giving the relatedness of those to each other in a table.

The figure 1 was changed as requested by aligning the two proximal laccases, and we modified the Materials and Methods, and the Results sections according to this change.

p. 7, l. 185: The supplementary material is missing. I hope that it would have been SDS-PAGE data.

The supplementary data are now available on the web site of the journal. SDS-PAGE data corresponds to Supporting information Figure 1.

p. 7, l. 191-192: The authors state that elimination of inhibiting factors led to increase in purification yield. No! These higher yields in activity are mainly due to removal of contaminating proteins and actual enrichment of the laccase, which can be seen in the increase of specific activity.

We cancelled this hypothesis in the revised version.

Figure 3: As already pointed out in my first review report, there are still not enough tickmarks on the temperature x-axis and information on the absolute activity in nkat units at 100% is still missing.

We added extra tickmarks in this figure and also in the following one to facilitate the reading of the data. In all the conditions, 100% relative activity was fixed in a range of 5 to 20 nkatal per mL of laccase activity, this information was added in Section 5.7.

p. 16, l. 363-366: As already pointed out in my first review report, You cannot state that Pestalotiopsis is secreting the laccases based on the presented results, because you only did recombinant expression in Aspergillus. It was never shown in this study that Pestalotiopsis is secreting these laccases.

Section 2.1 (before Table 2), we cancelled the term secreted as suggested.

This manuscript is a resubmission of an earlier submission. The following is a list of the peer review reports and author responses from that submission.

Round 1

Reviewer 1 Report

The present manuscript describes the cloning and expression of two different laccases from the fungus Pestalotiopsis sp, using A. niger as host microorganism. Their ability to perform biocatalysis at different pH, temperature and salt concentration was studied, as well as their kinetic properties and substrate specificity. Additionally, applicability of these enzymes to decolorize several commercial synthetic dyes was tested.

Expanding the biocatalysis toolbox available for industrial applications is always valuable. This goal is particularly appealing for enzymes used to degrade environmental pollutants from industrial processes to achieve more eco-friendly procedures. Hence, the results could be considered for publishing, although some issues need to be taken into consideration.

·    Introduction and the conclusions should be more concise and to the point, making the manuscript easier to read.

·    Line 39-42: The sentence “They are able to oxidize…” is not understandable.

·  Line 114 and also in the experimental section: it should be clear that the extinction coefficients of the proteins are theoretical, since there may be a difference between those and the experimental ones.

·    During the obtaining of both laccases from A. niger cultures at different salt concentrations, did this parameter affect the host microorganism growth, thus possibly influencing the expression of proteins?

·    Line 151: Table 1 refers to the kinetic parameters of the enzymes, not to their purification.

·   Line 152: it would be better to have the enzyme concentrations obtained per liter or ml of culture rather than the absolute amounts of protein.

·    Concerning the decolorization of commercial dyes:

-    Is there any particular reason behind choosing the ones selected for this experiment?

-    In the discussion, although the use of other laccases for the degradation of other dyes is described, there is no mention to the behaviour of other laccases towards the dyes tested in this research.

-    Additionally, dye degradation processes already published in the literature using either whole cells or immobilized cell free extract from Pestaloptiopsis are neither mentioned nor discussed.

Yanto et al (2014), Procedia Enviromental Sciences, 20, 235-244.

Nazareno-Saparrat and Hammer (2006), J. Basic Microbiology, 46(1), 28-33.

·    Surface charges of fungal laccases:

-   It is stated that an excess of negatively charged residues is a typical feature from marine enzymes several times in the manuscript, however there are not included any references to support this affirmation.

-     It is not specified in the Materials and Methods section which proteins are the “closest homologues of known structure available” used to predict the three-dimensional models, nor the level of homology they share.

·    The substrate concentration range used in the kinetic assays is not mentioned in the Materials and Methods section.

·    The conclusions section is missing.

·    Regarding to the Figures and Schemes:

-    Schemes of the reaction catalysed by laccases with and without mediator, as well as from the chemical structures of the compounds used as substrates would add valuable information for the reader.

-     Legends in Figures 4A and 4C are not readable.

-     There is a typo in the heading of plot B from Figure 5: an “a” from “Reactive Black” is missing.

-  Table 1: the number of digits should be reconsidered. The precision of the measurements probably does not motivate showing so many decimals.

One of the headings of the table is not written in bold.

-    Graphs obtained during the kinetic experiments performed should be included at least as supporting information.

·    Some typos found:

-   The space between the figure and the Celsius Degree symbol is missing throughout the manuscript.

-    Line 33: the fungus name is not in italics.

-    Line 424: Italics is missing in the name of the restriction enzymes

-    Line 459: subsection is missing in “(see section4.)”.

Reviewer 2 Report

The authors of the presented manuscript expressed and characterized two laccases from a marine fungus, which was isolated from a mangrove forest. In general, this a solid study and the characterizations were carried out thoroughly. I have only some minor comments:

p.2, l.69-70: As you write this sentence it sounds like Rhus vernicifera is also a fungus, which it is not. It is a tree!

Figure 1: In my opinion, this alignment is not so useful. The reader would be more interested in an alignment over the whole sequences including identity/homolgy of PsLac1/2 to other prominent laccases (eg. Myceliophthora thermophila, Trametes versicolor, Pleurotus ostreatus)

p.3, l.129: please show the sds-page in the supplement. What about glycosylation? Can you say somehting about that?

p.4, l.150-158: please prepare a purification table according to Burgess, R.R. Chapter 4 Preparing a Purification Summary Table. In Methods in Enzymology - Guide to Protein Purification; Burgess, R.R., Deutscher, M.P., Eds.; Elsevier, 2009; Vol. 463, pp. 29–34. Makes it much easier for the reader to find the interesting data.

Figure 3: 1. what is 100% rel. activity in units?, 2. Please put in also A,B,C,... into the graphs, 3. The tickmarks on the temperature x-axis are too few.

Figure 4: The legend in Figure 4A and C is not readable.

p.6, l.197/Table 1: The units in the text are not the same as in the table.
Don't put three or four digits after the comma.

Figure 7: Please define the red and blue colors in the figure. Pymol can do that.

p.9, l.270-273: You state that you report secretion of the ligninolytic enzymes by Pestalotiopsis, but you have only studied the heterologous expressed enzymes in Aspergillus. So, you cannot state this, although it is very likely that these enzymes are secreted in the original organism based on the fact that they have a signal peptide.

p.12,l.422: please add the reference for SignalP

p.14,l.485: reference Kelley, 2005 missing in the list.